# Syntrophic anaerobic photosynthesis via direct interspecies electron transfer

Phuc T. Ha[1], Stephen R. Lindemann[2], Liang Shi[3], Alice C. Dohnalkova[4], James K. Fredrickson[2], Michael T. Madigan[5] & Haluk Beyenal[1]

Microbial phototrophs, key primary producers on Earth, use $H_2O$, $H_2$, $H_2S$ and other reduced inorganic compounds as electron donors. Here we describe a form of metabolism linking anoxygenic photosynthesis to anaerobic respiration that we call 'syntrophic anaerobic photosynthesis'. We show that photoautotrophy in the green sulfur bacterium *Prosthecochloris aestaurii* can be driven by either electrons from a solid electrode or acetate oxidation via direct interspecies electron transfer from a heterotrophic partner bacterium, *Geobacter sulfurreducens*. Photosynthetic growth of *P. aestuarii* using reductant provided by either an electrode or syntrophy is robust and light-dependent. In contrast, *P. aestuarii* does not grow in co-culture with a *G. sulfurreducens* mutant lacking a trans-outer membrane porin-cytochrome protein complex required for direct intercellular electron transfer. Syntrophic anaerobic photosynthesis is therefore a carbon cycling process that could take place in anoxic environments. This process could be exploited for biotechnological applications, such as waste treatment and bioenergy production, using engineered phototrophic microbial communities.

[1] The Gene and Linda Voiland School of Chemical Engineering and Bioengineering, Washington State University, Pullman, Washington 99164, USA. [2] Biological Sciences Division, Pacific Northwest National Laboratory, Richland, Washington 99354, USA. [3] Department of Biological Sciences and Technology, School of Environmental Studies, China University of Geoscience, Wuhan, Hubei 430074, China. [4] Environmental Molecular Sciences Laboratory, Pacific Northwest National Laboratory, Richland, Washington 99354, USA. [5] Department of Microbiology, Southern Illinois University, Carbondale, Illinois 62901, USA. Correspondence and requests for materials should be addressed to H.B. (email: beyenal@wsu.edu).

Photosynthetic $CO_2$ fixation by cyanobacteria (oxygenic phototrophs), and purple and green sulfur bacteria (anoxygenic phototrophs) accounts for nearly one-half of global primary productivity[1]. Major electron donors for anoxygenic photosynthesis—an anaerobic process—include reduced inorganic compounds such as $H_2S$, $S^o$, $H_2$ and $Fe^{2+}$ [2]. In anoxic environments, electrons flow between microbes that form intimate syntrophic consortia to support complementary metabolisms. Perhaps the most intriguing of these electron flow mechanisms is the discovery of direct interspecies electron transfer, a process in which electrons are carried between cells through physical contact of conductive electron carriers[3,4]. Some examples include anaerobic ethanol oxidation by consortia composed of two *Geobacter* species[5] and the conversion of ethanol to methane by co-cultures of *Geobacter* species, and methanogenic Archaea such as *Methanosaeta*[6]. Although studies have suggested that direct interspecies electron transfer is common in anoxic methanogenic environments[3,4,7–9], whether related phenomena are more widespread, including in illuminated anoxic environments, is unknown.

Here we investigate this possibility and report the discovery of a new form of syntrophic metabolism, syntrophic anaerobic photosynthesis. We show that photoautotrophy in the green sulfur bacterium *Prosthecochloris aestaurii* can be driven by either electrons from a solid electrode or acetate oxidation via direct interspecies electron transfer from a heterotrophic partner bacterium, *Geobacter sulfurreducens*. Both interspecies electron transfer between *G. sulfurreducens* and *P. aestuarii*, and electron uptake by *P. aestuarii* are light-dependent processes. Light is required to drive photosynthetic metabolism in the phototroph. The discovery of syntrophic anaerobic photosynthesis reveals new possibilities for bioengineering phototrophic microbial communities for applications in the areas of waste treatment and bioenergy production.

## Results

**Electrode as electron donor for *P. aestuarii*.** Until now, the purple non-sulfur bacterium *Rhodopseudomonas palustris* TIE-1 has been the only phototroph known to be able to assimilate electrons from a solid electrode[10]. A few studies also suggested that some anaerobic phototrophs in a mixed culture are capable of using an electrode as an electron donor for photosynthesis[11,12]. However, neither the mechanisms nor the phototrophs responsible for electron exchange with electrodes have yet been identified, leaving an open question of whether other phototrophs can also capture electrons directly from a solid electrode for photosynthesis. In a previous study on electron transfer in a microbial mat harvested from Hot Lake—an episomitic lake in northern Washington (WA, USA)—we found the green sulfur bacterium *Prosthecochloris aestuarii* to predominate at the tip of a carbon microelectrode inserted into the illuminated mat[13]. In subsequent experiments, a pure culture of *P. aestuarii* was isolated. *P. aestuarii* can grow photoautotrophically utilizing sulfide or elemental sulfur ($S^o$) as an electron donor[14]. Recent studies have described several sulfur-oxidizing bacteria that are also capable of donating/accepting electrons from an electrode[15,16]. To determine whether the Hot Lake *P. aestuarii* can also utilize electrons from a solid electrode, we inoculated it into a sterile, two-chamber bioelectrochemical system containing growth medium in which $CO_2$ was the sole carbon source with it. The electrode was polarized at $-600\,mV_{Ag/AgCl}$ (slightly lower than the redox potential of $H_2S$ ($E^{o\prime}$ [$S^o/H_2S$] $-479\,mV_{Ag/AgCl}$)), and electron transfer from the electrode was assessed as cathodic current. A light-dependent cathodic current was detected that gradually increased over 24 h; after 17 days, the current reached

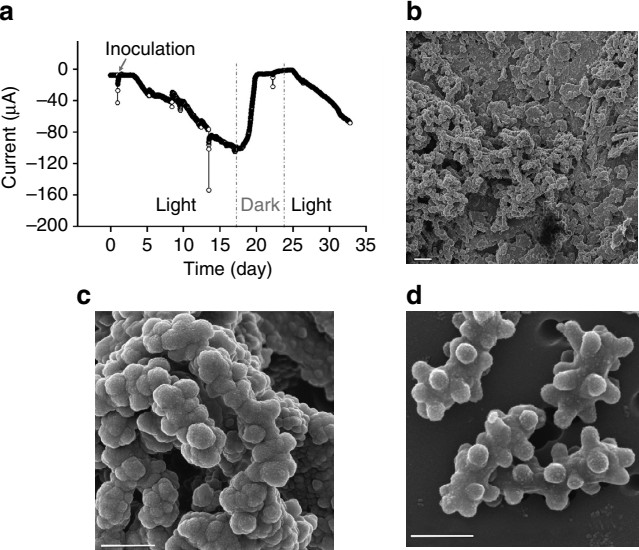

**Figure 1 | Solid electrode as electron donor for *P. aestuarii* growth.** (**a**) Cathodic current generated by a culture of *P. aestuarii* under light and dark conditions from a solid electrode polarized at $-600\,mV_{Ag/AgCl}$. Cathodic current was observed in all three biological replicates of experiments with this setup. (**b**) Scanning electron micrograph of the surface of the electrode (scale bar, 2 μm), showing the typical prosthecate morphology (**c**) of *P. aestuarii* cells (scale bar, 500 nm). Micrograph representative of 16 recorded images. (**d**) Scanning electron micrograph of typical sulfide-cultured *P. aestuarii* cells on an isopore membrane filter (scale bar, 500 nm). Micrograph representative of 27 recorded images.

104 μA ($\sim$36.74 μA cm$^{-2}$) (Fig. 1a). This electron uptake rate with a phototroph is well above that previously achieved with *R. palustris* TIE-1 (1.5 μA cm$^{-2}$)[10]. The continued development of cathodic current after medium exchanges indicates that no soluble electron shuttles are required for electron uptake by *P. aestuarii* (Supplementary Fig. 1a).

Scanning electron microscopy of a section of the electrode confirmed that the material accumulating was indeed a biofilm of *P. aestuarii* cells (Fig. 1b–d) in intimate contact with the electrode surface. These electrode-attached cells were used to inoculate fresh *P. aestuarii* growth medium containing sulfide as an electron donor, and growth occurred, confirming their viability on the electrode (Supplementary Fig. 1b).

To examine whether molecular hydrogen ($H_2$) mediated electron transfer between the electrode and *P. aestuarii*, we inoculated medium containing $H_2$ as the sole electron donor under illumination with *P. aestuarii*. *P. aestuarii* did not grow under these conditions, even after prolonged incubation ($>$10 days), indicating that this phototroph cannot use $H_2$ as an electron donor for photosynthesis (Supplementary Fig. 2a) and that uptake of electrons from the electrode was not mediated by $H_2$.

**Co-culture of *P. aestuarii* with *G. sulfurreducens*.** Bolstered by our findings with electrode-dependent photosynthesis by *P. aestuarii* (Fig. 1), we investigated the possibility that this organism could be grown in co-culture with a heterotrophic partner bacterium. In addition to transferring electrons to other bacterial species, the anaerobic and chemoheterotrophic bacterium *Geobacter sulfurreducens* can donate electrons to a solid electrode[17–19]. Since our experiments showed that *P. aestuarii* grew photoautotrophically using the electrons supplied by an electrode (Fig. 1), we hypothesized that *G. sulfurreducens* could also donate electrons to cells of

*P. aestuarii* to support the growth of both species under conditions in which neither could grow independently. To test this, we cultivated the two organisms in an illuminated, anoxic medium containing acetate but devoid of $HS^-$, $S^o$ or any other potential electron donors or acceptors. A small amount of thiosulfate (0.5 mM) was added as a source of sulfur for biosynthetic needs; however, thiosulfate is used neither as a photosynthetic electron donor by *P. aestuarii*[14,20] nor as an electron acceptor by *G. sulfurreducens*[21]. Since *P. aestuarii* is unable to fix $CO_2$ in the absence of $HS^-$ or $S^0$ (ref. 14) and *G. sulfurreducens* is unable to oxidize acetate in the absence of an electron acceptor, we reasoned that if a co-culture containing both organisms grew, it must be the result of interspecies electron transfer.

In these co-culture experiments, growth did indeed occur concomitantly with acetate consumption (Fig. 2a,b). Both *P. aestuarii* and *G. sulfurreducens* grew within the co-culture, as evidenced both by flow cytometry (Supplementary Fig. 3) and electron microscopy (Fig. 2c–e), in which the morphologically distinct species could be quantified. The TEM whole mount images of *P. aestuarii* (*P.a*) and *G. sulfurreducens* (*G.s*) co-cultures also displayed intimate cell connections, possibly functioning in electron transfers from *G. sulfurreducens* to *P. aestuarii* (Fig. 2d,e). The changes of acetate consumption and cell density when switching the co-cultures from dark to illuminated condition and vice versa (Fig. 3) suggest that the syntrophic growth between *G. sulfurreducens* and *P. aestuarii* is strictly light-dependent.

When these co-cultures were subcultured in fresh medium containing acetate, they continued to grow and consume acetate (Supplementary Fig. 4). This indicates that the syntrophic interaction was stable over multiple generations. The slower increase of biomass in subcultures than in original co-cultures is likely due to the cell density in the co-culture inoculum being lower than that in the two pure cultures. We conclude that growth of the co-culture represents a previously unrecognized

form of anoxygenic photosynthesis that exploits the complementary metabolisms of a heterotroph and a phototroph to the benefit of both. Because growth of the co-culture strictly depends on the metabolic activities of each partner bacterium, we term this process as 'syntrophic anaerobic photosynthesis'.

**Direct electron transfer supports syntrophic anaerobic photosynthesis.** Although our co-culture experiments revealed a metabolic interdependency between a chemotroph and a phototroph, the mechanism of electron transfer remained unclear. Chemically mediated interspecies electron transfer via exchange of a soluble organic compound such as formate, or of $H_2$, was a possible mechanism for the transfer of electrons from *G. sulfurreducens* to *P. aestuarii*. However, this was eliminated as a possibility in our co-cultures because *P. aestuarii* is unable to use $H_2$ or formate as an electron donor (Supplementary Fig. 2a). Three additional lines of evidence support our conclusion that the metabolic interdependency between *G. sulfurreducens* and *P. aestuarii* is not the result of a soluble electron carrier but instead requires cell–cell contact and direct interspecies electron transfer. First, the two organisms did not grow when cultured in the same medium but physically separated into two chambers by membrane filters (0.1-μm pore size), whereas a parallel culture containing both organisms in the same reactor grew (Supplementary Fig. 5). Second, in contrast to co-cultures with wild-type *G. sulfurreducens* under identical conditions, co-cultures of *P. aestuarii* and a *G. sulfurreducens* deletion mutant lacking the *ombB-omaB-omcB-orfS-ombC-omaC-omcC* gene cluster did not grow and acetate was not consumed (Fig. 4). The *ombB-omaB-omcB-orfS-ombC-omaC-omcC* gene cluster encodes a trans-outer membrane porin-cytochrome protein complex essential for extracellular electron transfer by *G. sulfurreducens* to solid electron acceptors such as ferrihydrite[22] or a solid electrode (Fig. 4a). Kanamycin (200 μg ml$^{-1}$) was added to the culture in

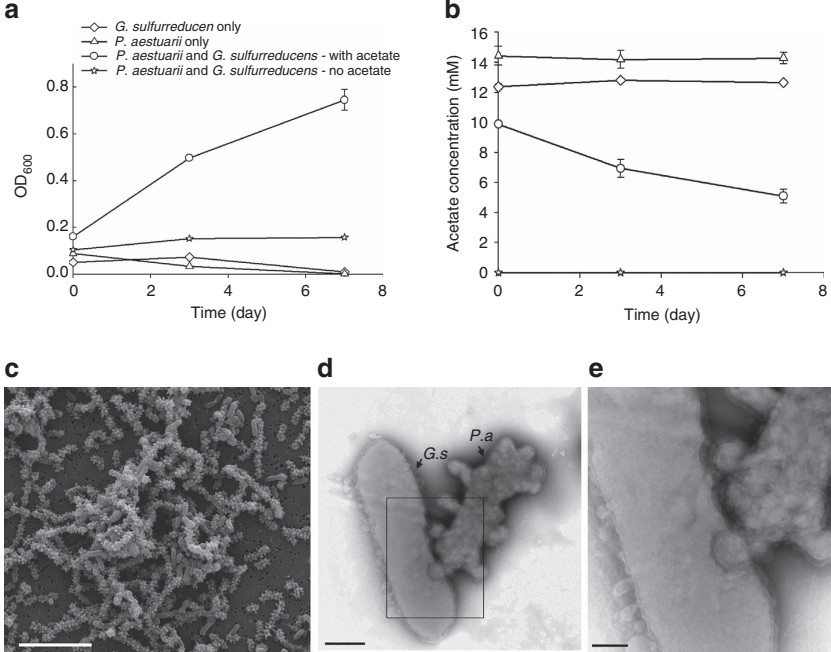

**Figure 2 | Syntrophic growth of *G. sulfurreducens* and *P. aestuarii*.** Variation in (**a**) cell density and (**b**) acetate consumption over time of *P. aestuarii* and *G. sulfurreducens* in axenic cultures and co-cultures. Each symbol is mean ± s.d. (*n* = 3). (**c**) SEM image of co-culture (scale bar, 5 μm). (**d,e**) TEM whole mount images of *G. sulfurreducens* (*G.s*) and *P. aestuarii* (*P.a*) co-cultures show two morphologically distinct species and their cell–cell contact via intimate extracellular associations (scale bars, 0.5 μm and 200 nm, respectively). These images are representative of the 20 SEM images and 26 TEM images obtained.

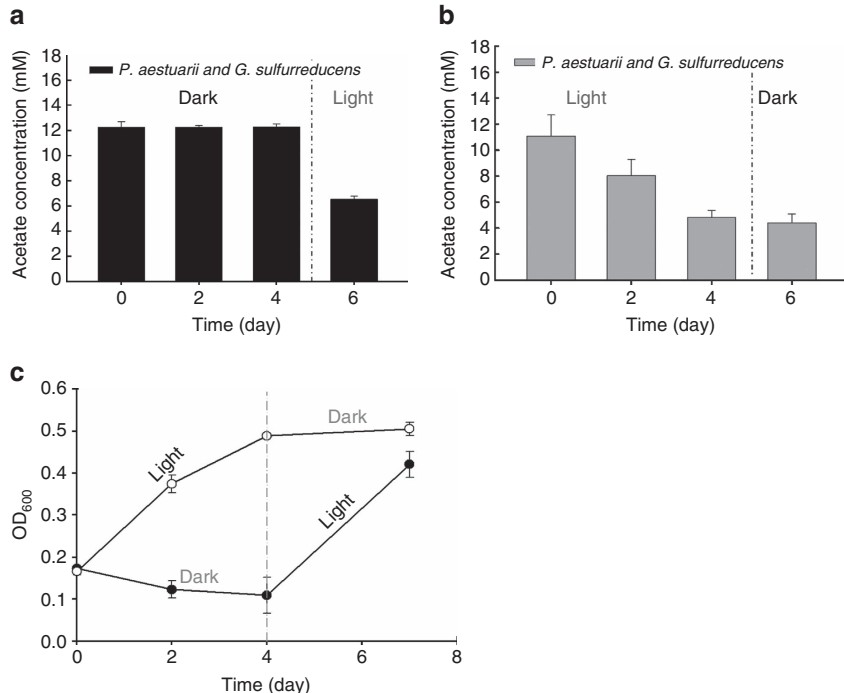

**Figure 3 | Variation in acetate concentration in the co-culture of *P. aestuarii* and *G. sulfurreducens* in the presence and absence of light.** (**a**) Switching the co-culture from a dark to an illuminated condition stimulated acetate consumption by the co-culture. (**b**) Switching from an illuminated to a dark condition stopped the photosynthesis of *P. aestuarii*, and therefore terminated acetate consumption by the co-culture. (**c**) Cell density ($OD_{600}$) of the co-culture of *P. aestuarii* and *G. sulfurreducens* in the presence and absence of light. The error bars represent the s.e.m. of replicated experiments ($n = 3$).

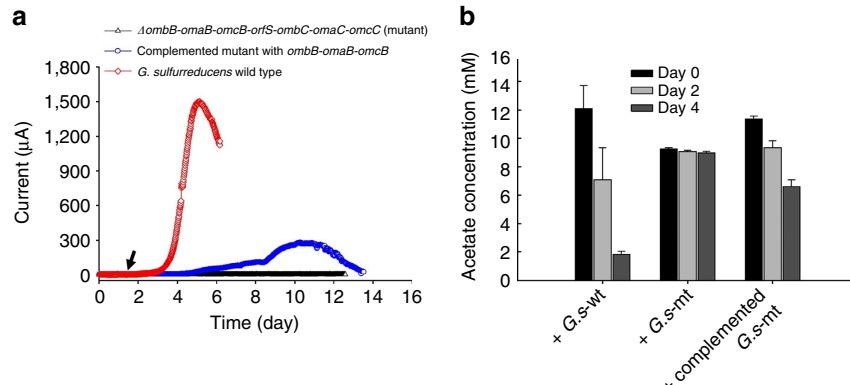

**Figure 4 | Current production and acetate consumption in co-culture by mutant and wild-type strains of *G. sulfurreducens*.** (**a**) The Δ*ombB-omaB-omcB-orfS-ombC-omaC-omcC* strain of *G. sulfurreducens* did not produce current on the electrode ($+300\,mV_{Ag/AgCl}$). The current was partially restored when it was complemented with the *ombB-omaB-omcB* gene. Three biological replicates were performed for each strain, and typical data are presented. The arrow shows the point where the reactors were inoculated. (**b**) Acetate consumption of co-cultures of *P. aestuarii* and wild-type *G. sulfurreducens* (*G.s*-wt); of *P. aestuarii* and Δ*ombB-omaB-omcB-orfS-ombC-omaC-omcC G. sulfurreducens* (*G.s*-mt), and of *P. aestuarii* and complemented Δ*ombB-omaB-omcB-orfS-ombC-omaC-omcC G. sulfurreducens* (complemented *G.s*-mt) which was introduced with the *ombB-omaB-omcB gene cluster*. The error bars represent the s.e.m. of replicate experiments ($n = 3$).

order to maintain the Δ*ombB-omaB-omcB-orfS-ombC-omaC-omcC* mutant of *G. sulfurreducens*; however, it did not affect the growth of this strain of *P. aestuarii*, which we found to be naturally kanamycin-resistant (Supplementary Fig. 2b). Third, when the *G. sulfurreducens* mutant was genetically complemented with the *ombB-omaB-omcB* gene cluster, which was previously shown to restore extracellular electron transfer capacity to $Fe^{3+}$ [22] and electrodes (Fig. 4a), and then tested in the

co-culture setup, light-dependent acetate consumption resumed (Fig. 4b).

*Geobacter* species require direct contact through outer membrane cytochromes and/or conductive pili for external electron transfer of electrons to insoluble electron acceptors (for example, electrode, Fe(III) oxide) and other organisms[23–25]. Electron micrographs of the co-culture showed intimate connections between cells of *G. sulfurreducens* and *P. aestaurii*

(Supplementary Fig. 6a,b). Connections included cell-to-cell contacts between the naturally protruding prosthecae of the phototroph and the cell surface and cell appendages of *G. sulfurreducens*. Moreover, haem-stained cell preparations from the co-culture showed an abundance of haem-stained filamentous structures that connected the cells (Supplementary Fig. 6c,d). The presence of these structures suggests that electron transfer between cells of *G. sulfurreducens* and *P. aestuarii* is mediated by haem-containing proteins. Collectively, these results support a direct interspecies electron transfer mechanism for syntrophic anaerobic photosynthesis.

## Discussion

Based upon these results, we propose a conceptual model for direct interspecies electron transfer between *G. sulfurreducens* and *P. aestuarii* in syntrophic anaerobic photosynthesis. Electrons originating from acetate are transferred from the heterotroph to the autotroph and used by the latter to fix $CO_2$ into cell material. The transfer benefits both organisms because it allows *G. sulfurreducens* to oxidize acetate as an electron donor and dispose of the electrons to an electron acceptor (the steps necessary to drive its bioenergetics), while simultaneously supplying the electrons needed by *P. aestaurii* to support photoautotrophic metabolism (Fig. 5). We demonstrated that *P. aestuarii* can obtain electrons from electrodes that were polarized at $-600\,mV_{Ag/AgCl}$. *G. sulfurreducens* is well known for its ability to transfer metabolic electrons directly to extracellular electron acceptors including electrodes[17,21,23,24,26]. Although the formal potential of the *G. sulfurreducens* biofilm grown on the surface of an electrode was shown to be around $-400\,mV_{Ag/AgCl}$ (refs 17,26–28), some studies have suggested that this bacterium can adjust its redox activity to the potential of the electron acceptor[12,18,19,29]. In addition, *G. sulfurreducens* is capable of reducing elemental sulfur[21], which is a known electron donor for *P. aestuarii*[14,20]. Thus, it is likely that in the co-culture the potential of *G. sulfurreducens* was lowered sufficiently for it to serve as a donor for *P. aestuarii* photosynthesis.

We here demonstrated the growth of a photoautotroph supported by direct electron transfer from a heterotrophic partner. Several recent studies point to syntrophic interspecies electron transfers between various anaerobes as a major means of heterotrophic carbon metabolism in anoxic microbial habitats[3,4,8,30]. Syntrophic anaerobic photosynthesis broadens this concept to include electron transfer from heterotrophs to phototrophs and reveals a previously unknown form of syntrophy that links anaerobic photosynthesis directly to anaerobic organic carbon metabolism. Moreover, although demonstrated here with the morphologically distinct green bacterium *Prosthecochloris*, we suspect that other species of green (and even purple) bacteria— organisms that are widely distributed in nature[2]—may also establish direct interspecies electron transfer relationships with electrogenic heterotrophic partners.

From an ecological perspective, syntrophic anaerobic photosynthesis could represent an important form of carbon metabolism in the anoxic zones of poorly mixed freshwater lakes, where sulfide limitations restrict the activities of anoxygenic phototrophs and a shortage of inorganic electron acceptors limits anaerobic respiration. In addition to the ecological importance of direct electron transfer between heterotrophs and phototrophs, the discovery of syntrophic anaerobic photosynthesis reveals new possibilities for bioengineering phototrophic microbial communities for applications in the areas of waste treatment and bioenergy production.

## Methods

**Bacterial strains and cultivation condition.** *Geobacter sulfurreducens* strain PCA (ATCC 51573) was used. The mutant strain of *G. sulfurreducens* in which the *ombB-omaB-omcB-orfS-ombC-omaC-omcC* gene cluster was replaced with an antibiotic-resistant gene and its complemented strain with the *ombB-omaB-omcB* gene cluster were created and characterized previously[22]. *Prosthecochloris aestuarii* strain 728 was isolated from phototrophic mat harvested from epsomitic, hypersaline Hot Lake near Oroville, WA, USA[31].

Strict anaerobic culturing procedures were used in serum vials sealed with butyl rubber stoppers throughout the study. *G. sulfurreducens* wild-type and mutant strains were cultured in a modified *Geobacter* medium containing acetate (10 mM) and fumarate (20 mM). The modified medium was determined to support the growth of *P. aestuarii* also when amended with sulfide and co-cultivation of both strains. The modified medium was made by dissolving 8.78 g of NaCl, 3.8 g of KCl, 0.25 g of NH₄Cl, 70 mg of CaCl₂ · 2H₂O, 0.5 g of KH₂PO₄, 0.5 g of MOPS, 1.0 g of Mg₂SO₄ · 7H₂O and 2.5 g of NaHCO₃ into 1 l of medium to which were added 1 × Wolfe's mineral solution and 1 × Wolfe's vitamin solution. The compositions of these trace mineral and vitamin solutions were previously described[32]. The medium was adjusted to pH 7.2, distributed into pressured vials, boiled and purged with $CO_2$:$N_2$ mixed gas (20%:80%) before being capped and sterilized with an autoclave. Thiosulfate (0.5 mM) was added for *P. aestuarii* biosynthesis to the media that were used in the cultivation of *P. aestuarii* on electrodes and in co-culture experiments. Kanamycin (200 µg ml⁻¹) was added to the media that were used for cultivation and co-culture experiments with *G. sulfurreducens* mutant strains[22].

*P. aestuarii* was routinely cultivated anaerobically in sulfide-containing medium. Sodium sulfide (3.5 mM as Na₂S · 9H₂O) was supplied to the *Geobacter* medium modified as described above. The sulfide-containing medium was allowed to sit in the dark for at least 2–3 h before being inoculated with *P. aestuarii*. The cultures were cultivated under constant illumination (8.3 ± 0.2 µmol m⁻²s⁻¹) and temperature (27 ± 0.4 °C) introduced by an incandescent light bulb (25 W).

Co-cultures of *G. sulfurreducens* and *P. aestuarii* were obtained from pure cultures that were in their stationary phase. Before being used to inoculate co-cultures, the supernatant from each pure culture (16 ml) was removed after centrifugation (4,200 g, 20 min). Cell pellets were then washed with 2 ml of substrate-free medium without sulfur or sulfide. The co-cultures were initiated with a 0.5-ml inoculum of washed *G. sulfurreducens* and a 0.5-ml inoculum of washed *P. aestuarii* added to 15 ml of modified *Geobacter* medium containing 10 mM of acetate as the sole electron source. Oxygen scavengers such as sodium sulfide or cysteine were omitted from the media. The co-cultures were also incubated under constant illumination (8.3 ± 0.2 µmol m⁻²s⁻¹) and temperature (27 ± 0.4 °C) introduced by an incandescent light bulb (25 W). In some experiments, the co-culture vials were wrapped with aluminium foil to exclude light.

The transferred co-cultures were initiated by inoculating fresh medium containing acetate as the sole electron donor with 1 ml of previously grown co-culture. The co-cultures were incubated under the same conditions as described above.

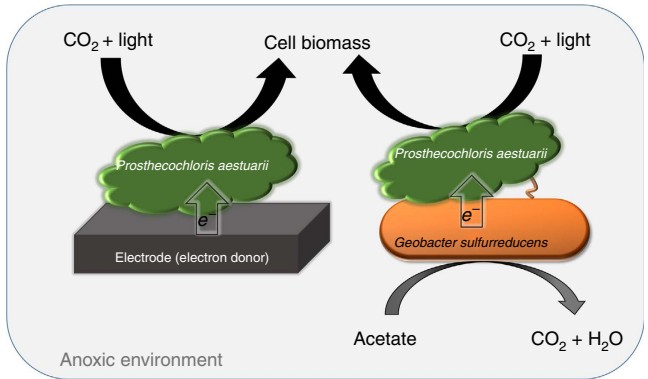

**Figure 5 | Conceptual model of syntrophic anaerobic photosynthesis of *G. sulfurreducens* and *P. aestuarii* via direct electron transfer.** Syntrophic growth of both species is supported through acetate oxidation by the heterotroph and $CO_2$ reduction by the phototroph.

**Analytical methods.** To determine the change in cell density during culture, 1 ml of culture medium was sampled during incubation for optical density (OD) measurement using a UV–vis spectrophotometer (Agilent, CA, USA). The OD of *G. sulfurreducens* and co-cultures were determined at a wavelength of 600 nm, while the OD of *P. aestuarii* cultures were monitored at 720 nm. The concentration of organic acids was determined using high-performance liquid chromatography (HPLC) equipped with an Aminex NPX-87H column (Bio-Rad, CA, USA). Sulfuric acid (5 mM) was used as eluent at 0.6 ml min⁻¹. Organic acids were detected at 210 nm using a UV detector (Agilent).

**Cell enumeration.** Cells in the co-cultures were sampled at various times during growth. Samples of 1 ml were centrifuged (11,300 g, 10 min), and the supernatants were discarded. Cell pellets were washed three times with 0.1 M phosphate buffer (PBS; pH 7.2) before being fixed in 4% paraformaldehyde. Flow cytometry was performed using a BD Influx Fluorescence-Activated Cell Sorter (BD Biosciences, San Jose, CA, USA). To disrupt cell aggregates, the cultures were treated with 100 mM $Na_2EDTA$ (Sigma Aldrich), passed through a 25-gauge needle 25 times, and then vortexed. The samples were then stained with SYBR Gold nucleic acid stain (ThermoFisher, Waltham, MA, USA). Optimization and calibration were performed before each flow cytometry analysis using 3.6-mm Ultra Rainbow fluorescent particles (Spherotech, Lake Forest, IL, USA). Forward and side scattering were used to gate out cellular debris, and the 488-nm argon laser was used to excite SYBR Gold while measuring emissions at 542/27 nm. Gating and median calculations for 20,000 cells were done using Flow Jo software (Tree Star, Ashland, OR). The ratios of the two distinct populations of cells within a mixed microbial community were identified from 20,000 recorded cells using size and complexity gates within FCS Express (Los Angeles, CA, USA) flow cytometry software.

**Electron microscopic analyses.** Scanning electron micrographs (SEM) provided imaging of the cell morphology, attachment to electrode surfaces and distribution within co-culture biofilms. To sample co-culture biofilms of G. sulfurreducens and P. aestuarii, a glass piece or a membrane filter as the biofilm substrate was placed into the serum vials before inoculation. The glass piece or membrane filter with attached biofilms was carefully removed and fixed for SEM, as were the graphite electrodes with cells attached from bioelectrochemical systems (BESs). Samples were fixed with 2% gluteraldehyde, 2% paraformaldehyde in 0.1 M PBS at pH 7.2 for up to 12 h at 4 °C, then washed three times in PBS for 10 min each, and dehydrated in ethanol series (10, 35, 50, 75, 95% and 3 × 100% for 10 min each). The samples were immediately immersed twice for 10 min in hexamethyldisilazane (Sigma Aldrich, MO, USA) followed by air-drying for 9–12 h in a fume hood. The samples were sputter-coated with gold and imaged with a Quanta SEM (FEI, Hillsboro, OR) and Orion Helium ion microscope (Zeiss, Peabody, MA).

TEM whole mounts were prepared by applying a 5-µl drop of cells from a co-culture onto a formvar-coated grid (Electron Microscopy Sciences (EMS, Hatfield, PA), and air-dried. The grids were viewed under an FEI Tecnai TEM equipped with a 200-keV LaB6 electron source (FEI, Hillsboro, OR). The detection of haem using thin-section TEM was performed with 3,3′-diaminobenzidine (DAB, EMS) treatment, following the previously described protocols[30,33]. Biofilms from co-culture were collected by carefully removing medium. The biofilm at the bottom of a vial was transferred to centrifuge tubes, centrifuged at low speed (850 g, 5 min) and fixed in 2% gluteraldehyde, 2% paraformaldehyde in 0.1 M PBS overnight at 4 °C. The fixative was replaced by three washes in PBS (10 min each) followed by two incubations (10 min each in the dark at room temperature) in PBS containing fresh DAB (0.05%). The haem stain was developed by the third incubation with PBS containing fresh DAB (0.05%) and 0.018% $H_2O_2$ (10 min in the dark at room temperature). Control samples received fresh DAB solution without $H_2O_2$. The reaction was stopped by washing three times in PBS before dehydration in an ascending series of ethanol and infiltration in LR White embedding resin (EMS) and cured at 55 °C for 24 h. The polymerized blocks were sectioned to 70-nm thin sections with a Leica Ultracut UCT ultramicrotome using a Diatome (Biel, Switzerland) diamond knife. The ultrathin sections were mounted on formvar-coated 100 mesh Cu grids sputtered with carbon. The TEM imaging was done using a Tecnai T-12 (FEI) with an LaB6 electron source, operating at 120 keV.

**Construction of bioelectrochemical systems and operational conditions.** The BESs for growing P. aestuarii were constructed as an H-type reactor consisting of two glass chambers (Adams & Chittenden Scientific Glass, CA, USA). The two chambers (100 ml working volume/each), which housed the counter and working electrodes, were separated by a cation-exchange membrane (5.07 cm$^2$). Counter and working electrodes were prepared by soaking $\frac{1}{4}$-inch-diameter rod-type graphite electrodes in 1 N HCl overnight, followed by sonication twice in 100% ethanol and twice in ultrapure DI water (30 min each). The electrodes were immersed in electrolyte, creating a consistent projected surface of ∼2.83 cm$^2$ for the working electrode and ∼4.3 cm$^2$ for the counter electrode. An Ag/AgCl reference electrode was inserted into the working chamber through a sampling port. The internal resistance of all BESs was <5 Ω. All the electrodes (working, reference, and counter electrodes) were electrically connected with potentiostat cables using alligator clips. Before inoculation, the chambers were filled with anaerobic, modified Geobacter medium as described above. To inoculate, 10 ml of grown P. aestuarii culture were centrifuged and the supernatant was discarded. The suspended cells were suspended in 2 ml of fresh medium and used to inoculate the 100-ml working chamber. The control experiment was conducted by filtering the supernatant from 10 ml of P. aestuarii culture through a membrane filter (0.22 µm) and introducing it into the working chamber. All experiments were conducted in an anaerobic chamber (to maintain an oxygen-free environment) containing an incandescent 25 W bulb with a constant temperature of 26 ± 0.7 °C. Dark cycle experiments were conducted by covering the BESs with a cardboard box to prevent light exposure. The working electrode was polarized at − 600 mV$_{Ag/AgCl}$ using a Gamry R300 (Gamry, PA, USA).

For testing the current production from the ΔombB-omaB-omcB-orfS-ombC-omaC-omcC mutant strain of G. sulfurreducens and the complemented strain into which the ombB-omaB-omcB gene cluster was reintroduced, we used 3-electrode flat-plate reactors (60 ml working volume). The working electrode was graphite (ground finish isomolded graphite plates, Glassmate grade GM-10, 25 mm × 25 mm × 3 mm, Poco Graphite, Inc., Decatur, TX). The experiments were conducted in an incubator (dark, 28 ± 0.5 °C) under continuous sparging with a $CO_2$:$N_2$ (20%:80%) gas mixture. The working electrode was polarized at 300 mV$_{Ag/AgCl}$ using a custom-built potentiostat[34]. The reactors were operated in batch mode with the modified Geobacter medium supplied with 10 mM acetate as the sole electron source.

**Operation of membrane-filter reactors.** A membrane-filter reactor was used for testing the growth of each of the species physically separated into different chambers but exchanging the same medium. The reactor was constructed using two vacuum filter holders (Millipore) connected by silicon tubing (Supplementary Fig. 5). A 0.1-µm pore size membrane filter was added to each filter holder such that the medium (100 ml) in each chamber was separately inoculated with a pure culture of a single bacterium. The outlet of each filter chamber was pumped (∼0.15 ml min$^{-1}$) to another one after being filtered a second time through another filter (0.45 µm). With this setup, the two bacteria were physically separated but the bulk solutions were well mixed to eliminate diffusion times between the G. sulfurreducens and P. aestuarii chambers.

After sterilization and addition of membrane filter, all reactors were moved to the anaerobic chamber and filled with the same modified Geobacter medium containing acetate as the sole electron donor. P. aestuarii and G. sulfurreducens from cultures previously grown with sulfide and acetate/fumarate, respectively, were harvested, washed and introduced into the reactor. The first chamber was inoculated with P. aestuarii, while the second chamber was inoculated with G. sulfurreducens. The acetate concentration in each chamber was monitored over time to assay the growth and activity of cells. For a control experiment, two bacteria were inoculated together. The acetate concentrations of samples from each chamber were monitored using HPLC (UV) as described above.

**Data availability.** All relevant data are available from the authors upon request.

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

## Acknowledgements

This research was supported by the Genomic Science Program (GSP) of the Office of Biological and Environmental Research (OBER), U.S. Department of Energy (DOE) and is a contribution of the Pacific Northwest National Laboratory (PNNL) Foundational Scientific Focus Area. Part of this research was performed using the Environmental Molecular Sciences Laboratory (EMSL), a national scientific user facility sponsored by the Department of Energy's Office of Biological and Environmental Research, located at the Pacific Northwest National Laboratory. We acknowledge the staff at the Franceschi Microscopy and Imaging Center (Washington State University, WA, USA) for their assistance and for providing the facilities for image analysis. We also thank William B. Chrisler (PNNL) for conducting the flow cytometry.

## Author contributions

P.T.H. designed and conducted the experiments and performed the analyses. L.S. created the *Geobacter sulfurreducens* mutant. M.T.M. isolated *Prosthecochloris aestuarii* from a microbial mat in Hot Lake (Washington, USA). A.C.D. conducted electron microscopy imaging. All authors contributed to experimental planning, data interpretation and writing of the manuscript.

## Additional information

**Competing financial interests:** The authors declare no competing financial interests.

