## [Peer Review File · Nature Communications]

Reviewers' comments:

Reviewer #1 (Remarks to the Author):

This manuscript describes the ability of *Prosthecochloris aestuarii* to accept electrons from electrodes or another microbial species to support anaerobic photosynthesis. It is becoming increasingly apparent that direct interspecies electron transfer (DIET) is an important form of syntrophy in anaerobic environments. This study with *P. aestuarii* is the first to describe DIET in which a phototroph is the electron-accepting partner. As noted in the manuscript, these results greatly expand the known microbial communities in which DIET may play an important role.

The study is very convincing with appropriate controls. I have no substantive criticisms.

Minor points:

Line 59. Incorrect reference. The correct reference for DIET with *Methanosaeta* is: Rotaru, et al., 2014. A new model for electron flow during anaerobic digestion: direct interspecies electron transfer to *Methanosaeta* for the reduction of carbon dioxide to methane. *Energy & Environ. Sci.* 7:408-415.

It is difficult to see the colors in the legend at the top of Figure 3A.

Reviewer #2 (Remarks to the Author):

This work provides further evidence that anoxygenic phototrophs can use electrons from other cells and surface in metabolism, and demonstrates an exciting new co-culture model for this process.

The key innovation appears to be the availability of a new model organism, *P. aestuarii*, which was isolated using sulfur as an electron donor, but came from from an electrode-based enrichment. There is evidence that this organism can accept electrons from an electrode in the presence of light, and that this organism can also accept electrons from *G. sulfurreducens* in the presence of light. As *P. aestuarii* cannot use hydrogen, formate, or acetate, the most common sources confounding this kind of study are unlikely, and most controls support the central idea of the co-culture.

The electrochemistry data for *P. aestuarii* is weaker, as is the electrochemical data for a knockout of *Geobacter* that is supposed to aid the model. Lacking an explanation for odd behavior in these figures (1B and 3A) removing this work does not alter the conclusions. The most important observations show the co-culture consuming acetate and increasing in OD in the light, and this not happening in the dark (but is buried in supplemental). Lacking isotope data proving CO₂ uptake, or mutants in *P. aestuarii*, all the other observations are interesting but not central. Even the idea of a shuttle is supported by non-evidence rather than a positive experiment showing how it would behave if it were to occur.

Specific comments

The CV data in particular, is disappointing. This is potentially one of the first new cathodic electrochemically active bacteria, and it appears to be an n=1 experiment showing one round of light/dark growth and a CV at fast scan rate. This could have added an essential piece of evidence to support the hypothesis of co-culture growth with *G. sulfurreducens*, in the form of onset potential(s) for when cathodic electrons are consumed that reconcile with known potentials produced by *G. sulfurreducens*. By performing the sweep at 10 mV/s, the capacitance is large and we get a poor to unclear sense of the thermodynamics of this process. Worse, the reverse

sweep for the 'fresh medium' control looks similar to the 'light' experiment, and there appear to be redox-active elements in the system, even in the cell-free medium. As it appears unreplicated, removing the data is best --and in the future suggest much slower scans, smaller electrodes, or more baseline controls

Fig 1. Please show and discuss data in current per unit area ($\mu\text{A}/\text{cm}^2$) - this allows comparison with past and future work. For example, $36 \mu\text{A}/\text{cm}^2$ with a phototroph is well above anything achieved previously and greatly adds to its significance.

Fig 1. If this is 'typical' of *P. aestuarii*, please show a typical cell from pure culture for comparison.

If this is an H-cell like setup, please provide estimates of the junction potentials and i-R drop from the reference electrode in the methods - these values can vary greatly depending on the geometry of the reactors and methods used

Please explain in the text (e.g., around line 135 where Ex 5 is mentioned) why in some cases five days of growth produced an OD of 0.6 in co-cultures, while in others that took 10 days of growth. What bothers me about these differences is what should someone expect if they repeat this - the authors say growth is "robust"...but there is no indication anywhere of doubling times or yields to support this statement. What is "normal" or what is the variation between experiments?

I appreciate what the 3-chamber separated by filters experiment is supposed to test, but with how slow diffusion is, and what is known about separating metals by barriers to test shuttling (e.g. AQDS will not reduce Fe(III) in a dialysis bag, even through the pores are small enough), there is no way this experiment should have ever produced growth when two bacteria are separated by two filters and a chamber. The only way this, or any hypothesis of shuttling, could be supported is if an actual shuttle was added and growth occurred.

Fig 3a The wild type appears to take over 3 days to initiate growth. The number of replicates of each experiment is not given, and the final density reached by the wild type is not clear (if it is based on the previous sized electrodes, this is a very poor *G. sulfurreducens* result). If this were a paper about *G. sulfurreducens* BES reactors, I would not accept this data. I suggest just taking it out and using Fig 3b, which has replicates and speaks to the hypothesis.

Fig extended 4: why so much fundamental data proving syntrophic growth is hidden in supplementary data (such as 4abc), while an odd mutant experiment without light/dark controls (Fig 3b) is shown is odd. Please show ex Fig 4 in the paper to make the case for the actual physiology being discovered.

The discussion of the 'local potential' of *G. sulfurreducens* around line 200 is weak, and lacking a good indication of what potential is required to drive *P. aestuarii*, this is just guessing. There are plenty of actual CV's of *G. sulfurreducens* wild type and mutant cells showing how the potential of this organism's anodic electron transfer changes that are not cited.

Data that can be removed;

-Fig 1b is at such a high scan rate and has so many questionable regions it does not help the argument.

-Fig 3 is meaningless, as hydrogen is not a donor for the cell, and as similar attempts to measure hydrogen in sterile systems later turned out to be meaningless when formate dehydrogenases/hydrogenases released by cells were found to create the hydrogen.

-Fig Ex 7 will never work, diffusion is too slow to support a whole culture through two filters and medium.

Reviewer #3 (Remarks to the Author):

The manuscript of Ha et al describes a previously unrecognized process of a direct electron transfer from a chemoheterotroph to a photoautotroph, which authors named "syntrophic anaerobic photosynthesis". Using various techniques including electron microscopy, various growth tests and experiments with *Geobacter* mutants lacking a protein complex responsible for extracellular electron transport, Ha et al could show that electron transport proceeds through a direct contact between *Geobacter* and *Prosthecochloris* cells rather than via H₂ or formate. In this case, *Geobacter* cells functioned as a cathode, and a cathode could indeed substitute them.

The work is convincing, well-written and carefully done. It exploits a repertoire of known microbial growth modes and represents an important advantage in microbial physiology.

Minor comments.

Line 52: "Fe+2": "Fe²⁺"

Lines 136-138 and 179-182: redundant

Lines 275-277: please provide details for the HPLC tests (gradient, flow rate, retention times)

Please consider integration of Extended Data Figures in the main text. On the contrary, Fig. 4 is not particularly informative for being a stand-alone figure.

How the purity of the culture was controlled in the experiments with electrodes?

Kanamycin was added to the media that were used for cultivation and co-culture experiments with *G. sulfurreducens* mutant strains (lines 250-251). Please provide evidence that *Prosthecochloris* is resistant to kanamycin.

Is the mode of growth of the green sulfur bacterium in association with *Geobacter* mixotrophy or autotrophy?

Reviewer #1 (Remarks to the Author):

This manuscript describes the ability of *Prosthecochloris aestuarii* to accept electrons from electrodes or another microbial species to support anaerobic photosynthesis. It is becoming increasingly apparent that direct interspecies electron transfer (DIET) is an important form of syntrophy in anaerobic environments. This study with *P. aestuarii* is the first to describe DIET in which a phototroph is the electron-accepting partner. As noted in the manuscript, these results greatly expand the known microbial communities in which DIET may play an important role.

The study is very convincing with appropriate controls. I have no substantive criticisms.

Minor points:

Line 59. Incorrect reference. The correct reference for DIET with *Methanosaeta* is: Rotaru, et al., 2014. A new model for electron flow during anaerobic digestion: direct interspecies electron transfer to *Methanosaeta* for the reduction of carbon dioxide to methane. *Energy & Environ. Sci.* 7:408-415.

We thank the reviewer for noticing this. We have corrected the reference.

It is difficult to see the colors in the legend at the top of Figure 3A.

We have modified Figure 2 to make the colors in the legend clear. Figure 3 (in the original MS) has now become Figure 4a after revision.

Reviewer #2 (Remarks to the Author):

This work provides further evidence that anoxygenic phototrophs can use electrons from other cells and surface in metabolism, and demonstrates an exciting new co-culture model for this process.

The key innovation appears to be the availability of a new model organism, *P. aestuarii*, which was isolated using sulfur as an electron donor, but came from from an electrode-based enrichment. There is evidence that this organism can accept electrons from an electrode in the presence of light, and that this organism can also accept electrons from *G. sulfurreducens* in the presence of light. As *P. aestuarii* cannot use hydrogen, formate, or acetate, the most common sources confounding this kind of study are unlikely, and most controls support the central idea of the co-culture.

The electrochemistry data for *P. aestuarii* is weaker, as is the electrochemical data for a knockout of *Geobacter* that is supposed to aid the model. Lacking an explanation for odd behavior in these figures (1B and 3A) removing this work does not alter the conclusions. The most important observations show the co-culture consuming acetate and increasing in OD in the light, and this not happening in the dark (but is buried in supplemental). Lacking isotope data proving CO₂

uptake, or mutants in *P. aestuarii*, all the other observations are interesting but not central. Even the idea of a shuttle is supported by non-evidence rather than a positive experiment showing how it would behave if it were to occur.

We thank the reviewer for his/her helpful comments, which improved the manuscript. We have considered all the suggestions critically and revised the manuscript accordingly. Since *P. aestuarii* did not use acetate, the only C source in the medium was CO₂. Thus if *P. aestuarii* grew, which it did, it had to be fixing CO₂. As the reviewer suggested, the isotopic data proving CO₂ uptake, or mutants in *P. aestuarii*, were not central.

Specific comments

The CV data in particular, is disappointing. This is potentially one of the first new cathodic electrochemically active bacteria, and it appears to be an n=1 experiment showing one round of light/dark growth and a CV at fast scan rate. This could have added an essential piece of evidence to support the hypothesis of co-culture growth with *G. sulfurreducens*, in the form of onset potential(s) for when cathodic electrons are consumed that reconcile with known potentials produced by *G. sulfurreducens*. By performing the sweep at 10 mV/s, the capacitance is large and we get a poor to unclear sense of the thermodynamics of this process. Worse, the reverse sweep for the 'fresh medium' control looks similar to the 'light' experiment, and there appear to be redox-active elements in the system, even in the cell-free medium. As it appears un-replicated, removing the data is best --and in the future suggest much slower scans, smaller electrodes, or more baseline controls

We agree with the reviewer's suggestion. We have removed Figure 1B as well as the related discussion, which did not alter the conclusions.

Fig 1. Please show and discuss data in current per unit area ($\mu\text{A}/\text{cm}^2$) - this allows comparison with past and future work. For example, 36 $\mu\text{A}/\text{cm}^2$ with a phototroph is well above anything achieved previously and greatly adds to its significance.

We have added a discussion on lines 83-84 (page 4) which compares our electron uptake rate with that of the previous cathodic phototroph, *R. palustris* TIE-1, presented by Bose et al., 2014.

Fig 1. If this is 'typical' of *P. aestuarii*, please show a typical cell from pure culture for comparison.

We have added a SEM of typical *P. aestuarii* cells from pure culture to Figure 1 (Figure 1d).

If this is an H-cell like setup, please provide estimates of the junction potentials and i-R drop from the reference electrode in the methods - these values can vary greatly depending on the geometry of the reactors and methods used

Both chambers were filled with the same medium so that the junction potential was eliminated. The detail estimated values for our H-type reactors were added in the materials and methods section (Line 336-344, Page 18-19).

Please explain in the text (e.g., around line 135 where Ex 5 is mentioned) why in some cases five days of growth produced an OD of 0.6 in co-cultures, while in others that took 10 days of growth. What bothers me about these differences is what should someone expect if they repeat this - the authors say growth is "robust"...but there is no indication anywhere of doubling times

or yields to support this statement. What is "normal" or what is the variation between experiments?

We have added text on lines 147-149 (page 9) to explain the difference of the growth time obtained in the co-cultures in Extended Data Fig. 6 (up to 10 days) with that of the others (only 5 days).

The co-cultures in Extended Data Fig. 6 were **transferred-cultures** of *P. aestuarii* and *G. sulfurreducens*. The subcultured co-cultures were made by inoculating fresh medium (15 mL) containing acetate as the sole electron donor with 1 mL of previously grown co-culture ($OD_{600} \sim 0.6$). Therefore, they initially had fewer cells than the initial co-culture obtained by mixing 0.5 mL of two pure cultures (*P. aestuarii* and *G. sulfurreducens*), which were more concentrated cell suspensions. Detailed information is given in the Materials and Methods section (Lines 263-275, pages 15-16).

I appreciate what the 3-chamber separated by filters experiment is supposed to test, but with how slow diffusion is, and what is known about separating metals by barriers to test shuttling (e.g. AQDS will not reduce Fe(III) in a dialysis bag, even though the pores are small enough), there is no way this experiment should have ever produced growth when two bacteria are separated by two filters and a chamber. The only way this, or any hypothesis of shuttling, could be supported is if an actual shuttle was added and growth occurred.

We appreciate this comment by the reviewer. We agree that a long diffusion time could prevent the growth of the co-culture. Therefore, we revised our reactor design (Extended Data Figure 5 (top); see below) and reran these experiments. This time we used filters to grow cultures and recirculate medium. Membrane filters (0.1- μm pore size) were added to each chamber which was inoculated with a pure culture of a single bacterium. The effluent of each filter chamber was pumped to another one after being filtered a second time through an attached syringe filter. With this setup, the two bacteria were physically separated but the bulk media of the *G. sulfurreducens* chamber and the *P. aestuarii* chamber were well mixed and solutes could freely exchange through the filter. This design eliminated or greatly reduced the diffusion limitation, and if one of the organisms had generated a mediator it would have been rapidly transported to the other chamber. In a separate control experiment, when the two bacteria were incubated together, acetate was consumed and growth observed. We have updated the Methods section to describe this new setup (lines 365-381, page 20). We have also updated the Extended Data Figure 7 (original MS) with the schematic diagram (below) and the results from this new filter reactor experiment. This figure is now Extended Data Figure 5. Overall, we observed no significant change in total acetate concentration, which provides further evidence that no soluble electron shuttles were involved in transferring electrons between the two strains.

Figure 5 (Revised MS) Illustrated diagram of membrane filter chambers used to examine whether soluble electron shuttle(s) could be involved in the interspecies electron transfer between *P. aestuarii* and *G. sulfurreducens*. The two species were physically separated into two separated chambers in which membrane filters (0.1-mm pore size) were inserted at the bottom. The effluent of each filter chamber was pumped (~0.15 ml/min) to another one after being filtered a second time through an attached syringe filter.

Fig 3a The wild type appears to take over 3 days to initiate growth. The number of replicates of each experiment is not given, and the final density reached by the wild type is not clear (if it is based on the previous sized electrodes, this is a very poor *G. sulfurreducens* result). If this were a paper about *G. sulfurreducens* BES reactors, I would not accept this data. I suggest just taking it out and using Fig 3b, which has replicates and speaks to the hypothesis.

We have updated this Figure 3a (which is now Figure 4a) with the results from longer-term cultures for all *G. sulfurreducens* strains in flat-plate reactors. The figure shows typical current development with the maximum value reached by the wild type and mutants in each batch. The number of replicates is also listed in the updated caption.

We also updated the Methods section related to this experiment (line 358-364, page 19-20)

Fig extended 4: why so much fundamental data proving syntrophic growth is hidden in supplementary data (such as 4abc), while an odd mutant experiment without light/dark controls (Fig 3b) is shown is odd. Please show ex Fig 4 in the paper to make the case for the actual physiology being discovered.

We agree with these comments by the reviewer that the results shown in Extended Data Figure 4 are critical. We have moved Extended Data Figure 4 to the main text as Figure 3.

The discussion of the 'local potential' of *G. sulfurreducens* around line 200 is weak, and lacking a good indication of what potential is required to drive *P. aestuarii*, this is just guessing. There are

plenty of actual CV's of *G. sulfurreducens* wild type and mutant cells showing how the potential of this organism's anodic electron transfer changes that are not cited.

We have revised this part of our discussion, which is now accompanied by additional supporting references (Line 204-214, page 12).

Data that can be removed;

-Fig 1b is at such a high scan rate and has so many questionable regions it does not help the argument.

The data have been removed as suggested.

-Fig 3 is meaningless, as hydrogen is not a donor for the cell, and as similar attempts to measure hydrogen in sterile systems later turned out to be meaningless when formate dehydrogenases/hydrogenases released by cells were found to create the hydrogen.

Figure 3 shows that the co-culture did not grow if the external electron transfer ability of *G. sulfurreducens* was completely knocked out. This supports the central model that strictly requires external electron transfer ability through heme-containing proteins of *G. sulfurreducens*. This model is further supported by the electron micrographs in Extended Data Figure 8c and d (which are now Extended Data Figure 6c and d). These show an abundance of heme-stained filamentous structures that connected the cells. Therefore, we believe it is critical to retain these data in the revised manuscript. However, as the reviewer suggested, we have removed the data on the measurement of hydrogen near the electrode surface.

-Fig Ex 7 will never work, diffusion is too slow to support a whole culture through two filters and medium.

We updated the experimental setup to eliminate the slow diffusion, as addressed above.

Reviewer #3 (Remarks to the Author):

The manuscript of Ha et al describes a previously unrecognized process of a direct electron transfer from a chemoheterotroph to a photoautotroph, which authors named "syntrophic anaerobic photosynthesis". Using various techniques including electron microscopy, various growth tests and experiments with *Geobacter* mutants lacking a protein complex responsible for extracellular electron transport, Ha et al could show that electron transport proceeds through a direct contact between *Geobacter* and *Prosthecochloris* cells rather than via H₂ or formate. In this case, *Geobacter* cells functioned as a cathode, and a cathode could indeed substitute them.

The work is convincing, well-written and carefully done. It exploits a repertoire of known microbial growth modes and represents an important advantage in microbial physiology.

Minor comments.

Line 52: "Fe+2": "Fe2+"
Corrected.

Lines 136-138 and 179-182: redundant

The first sentence (original lines 136-138, which is now 151-153) proposes syntrophic anaerobic photosynthesis. The second sentence (original lines 179-182, which is now 187-190) mentions that syntrophic anaerobic photosynthesis occurs through direct electron transfer between two strains. We prefer to keep these sentences in the revised manuscript.

Lines 275-277: please provide details for the HPLC tests (gradient, flow rate, retention times)
Detailed methods for HPLC analyses are now provided.

Please consider integration of Extended Data Figures in the main text. On the contrary, Fig. 4 is not particularly informative for being a stand-alone figure.

This comment is the same as a suggestion by reviewer #2. We have moved this Extended Fig. 4 to the main text as Figure 3.

How the purity of the culture was controlled in the experiments with electrodes?

SEM imaging was used to check whether there was contamination on electrodes after operation. In addition, in some electrode experiments with mutants, kanamycin was provided continuously, which helped to maintain the mutants and to prevent contamination.

Kanamycin was added to the media that were used for cultivation and co-culture experiments with *G. sulfurreducens* mutant strains (lines 250-251). Please provide evidence that *Prosheochloris* is resistant to kanamycin.

Evidence that *P. aestuarii* is naturally resistant to kanamycin and can grow in medium containing kanamycin (200 mg/ml) is provided in Extended Data Figure 2b. We have added a sentence (line 173-176, page 10) stating that *P. aestuarii* growth was not inhibited by kanamycin at 200 µg/mL.

Is the mode of growth of the green sulfur bacterium in association with *Geobacter* mixotrophy or autotrophy?

In this co-culture, the *P. aestuarii* used electrons supplied by *G. sulfurreducens* (from the oxidation of acetate) to fix CO₂ (in the medium and gas phase). In its pure culture, *P. aestuarii* cannot consume acetate. Therefore, the mode of growth of the *P. aestuarii*-*G. sulfurreducens* co-culture is autotrophy.

REVIEWERS' COMMENTS:

Reviewer #2 (Remarks to the Author):

At the end of the day, this new demonstration of interspecies electron transfer, between a metal-reducing bacterium and a phototroph, is an important contribution. Absolute proof of direct vs. shuttles, or the molecular mechanism, will take many additional experiments in both organisms, and is not the point of the article. I suspect most in the field will want to read this article and try their hand growing this cool bacterium.

I appreciate the new experiments aimed at testing the shuttling hypotheses and clarifying experimental details. I am still wary of the long lag times, such as in the growth experiments on the electrodes in Figure 4, the lack of empty vector controls which would be standard in a good microbiology journal, and the use of kanamycin in electrode-based experiments which is established to prevent robust electrode growth. I assume other readers will also see these experiments as less well controlled, and weigh them appropriately.

Reviewer #3 (Remarks to the Author):

All comments have been addressed

REVIEWERS' COMMENTS:

Reviewer #2 (Remarks to the Author):

At the end of the day, this new demonstration of interspecies electron transfer, between a metal-reducing bacterium and a phototroph, is an important contribution. Absolute proof of direct vs. shuttles, or the molecular mechanism, will take many additional experiments in both organisms, and is not the point of the article. I suspect most in the field will want to read this article and try their hand growing this cool bacterium.

I appreciate the new experiments aimed at testing the shuttling hypotheses and clarifying experimental details. I am still wary of the long lag times, such as in the growth experiments on the electrodes in Figure 4, the lack of empty vector controls which would be standard in a good microbiology journal, and the use of kanamycin in electrode-based experiments which is established to prevent robust electrode growth. I assume other readers will also see these experiments as less well controlled, and weigh them appropriately.

We appreciate these comments. Our manuscript focused on the demonstration of interspecies electron transfer, between a metal-reducing bacterium and a phototroph. The molecular mechanism is not the point of this study. We believe that many researchers will be interested to identify the mechanism using different tools than us. Such as, one of a recent published paper had suggested that the direct contact between cells and elemental sulfur is required for green sulfur bacterial growth (Hanson et al., 2016). This new model for sulfur oxidation by green sulfur bacteria, which have been largely unknown, also support our central finding in this manuscript.

We had performed additional experiments to address previous reviewer's suggestions. Empty vector controls are not critically needed for our research and conclusion. We can try to do this but it will take another year and we don't see that this would change or affect our conclusion and our manuscript.

The reviewer concerned that long lag times in the growth experiments of *Geobacter* wild type and mutants on electrode. It should be noted that the data presented on Figure 4a included the first ~1.5 days of current generated under cell-free condition to have a base line. We added an arrow to Figure 4a to show inoculation point and introduced text to the figure caption. We thank the reviewer for noticing this. Operating the cells for 1-2 days without inoculating is important for us to assure sterility of the system and generate a baseline data.

Kanamycin were used in some experiments which used *Geobacter* mutants, including cell growing on electrode and co-culture experiment. This is required to maintain the mutated genes as it was described in literatures (Liu et al., 2014)

References:

- Hanson TE, Bonsu E, Tuerk A, Marnocha CL, Powell DH, Chan CS. 2016. *Chlorobaculum tepidum* growth on biogenic S(0) as the sole photosynthetic electron donor.. Environ Microbiol. 9:2856-67.
- Liu Y, Wang Z, Liu J, Levar C, Edwards MJ, Babauta JT, Kennedy DW, Shi Z, Beyenal H, Bond DR, Clarke TA, Butt JN, Richardson DJ, Rosso KM, Zachara JM, Fredrickson JK, Shi

L. 2014. A trans-outer membrane porin-cytochrome protein complex for extracellular electron transfer by *Geobactersulfurreducens* PCA. Environ Microbiol Rep. 6(6): 776–7

Reviewer #3 (Remarks to the Author):

All comments have been addressed

No response is required.